# Racial Disparity in Pregnancy Risks and Complications in the US: Temporal Changes during 2007–2018

**DOI:** 10.3390/jcm9051414

**Published:** 2020-05-10

**Authors:** Eran Bornstein, Yael Eliner, Frank A. Chervenak, Amos Grünebaum

**Affiliations:** 1Division of Maternal-Fetal Medicine, Department of Obstetrics and Gynecology, Lenox Hill Hospital—Northwell Health/Zucker School of Medicine, New York, NY 10075, USA; fcervenak@northwell.edu (F.A.C.); amos.grunebaum@gmail.com (A.G.); 2School of Public Health, Boston University, Boston, MA 02118, USA; yaeleliner@gmail.com

**Keywords:** racial disparity, race, ethnicity, pregnancy risk factors, pregnancy complications

## Abstract

Maternal race and ethnicity have been associated with differences in pregnancy related morbidity and mortality. We aimed to evaluate the trends of several pregnancy risk factors/complications among different maternal racial/ethnic groups in the US between 2007 and 2018. Specifically, we used the Center for Disease Control and Prevention (CDC) natality files for these years to assess the trends of hypertensive disorders of pregnancy (HDP), chronic hypertension (CH), diabetes mellitus (DM), advanced maternal age (AMA) and grand multiparity (GM) among non-Hispanic Whites, non-Hispanic Blacks and Hispanics. We find that the prevalence of all of these risk factors/complications increased significantly across all racial/ethnic groups from 2007 to 2018. In particular, Hispanic women exhibited the highest increase, followed by non-Hispanic Black women, in the prevalence of HDP, CH, DM and AMA. However, throughout the entire period, the overall prevalence remained highest among non-Hispanic Blacks for HDP, CH and GM, among Hispanics for DM, and among non-Hispanic Whites for AMA. Our results point to significant racial/ethnic differences in the overall prevalence, as well as the temporal changes in the prevalence, of these pregnancy risk factors/complications during the 2007–2018 period. These findings could potentially contribute to our understanding of the observed racial/ethnic differences in maternal morbidity and mortality.

## 1. Introduction

Racial and ethnic differences in obstetric, perinatal and neonatal outcomes in the United States (US) are well-documented [1,2,3]. Such differences have been shown to impact both maternal and neonatal morbidity and mortality. For example, non-Hispanic Black women have been reported to be several times more likely than non-Hispanic White and Hispanic women to die from a pregnancy related cause [1,2,3,4], as well as to have a preventable death [5,6]. For example, in 2018, the US maternal mortality rate for Black women was 37.1 per 100,000 live births, whereas the rates for White and Hispanic women were 14.7 and 11.8 per 100,000 live births, respectively [7].

While maternal mortality has traditionally been the focal point of research on pregnancy related racial and ethnic inequalities, the differences leading to this adverse outcome likely originate earlier in the pregnancy. Several studies have found significant differences in pregnancy related risks and complications among different maternal racial/ethnic groups. For example, Black women were found to be more likely than White women to suffer from pregnancy induced hypertension, chronic hypertension, preterm labor, antepartum and postpartum hemorrhage and peripartum infections [8,9,10,11,12,13], while Hispanic women were found to be at greater risk than White women for gestational diabetes, peripartum infections and postpartum hemorrhage [8,10,11,12,14]. Likewise, Creanga et al. (2014) reported that the rate of severe maternal morbidity at delivery or during the postpartum period for non-Hispanic Blacks was 2.1 times the rate for non-Hispanic Whites, and that the rate for Hispanics was 1.3 times the rate for non-Hispanic Whites [15].

In this study, we aimed to understand the trends of several pregnancy risk factors and complications—hypertensive disorders of pregnancy (HDP), chronic hypertension (CH), diabetes mellitus (DM), advanced maternal age (AMA) and grand multiparity (GM)—among US gravidas from different racial/ethnic groups in the time period between 2007 and 2018. Data regarding temporal changes in these pregnancy related risk factors/complications among the different racial/ethnic groups have the potential to clarify the root causes of the observed racial/ethnic differences in maternal morbidity and mortality and to shed light on the wide array of suggested explanations.

## 2. Materials and Methods

This is a retrospective cohort study utilizing the Center for Disease Control and Prevention (CDC) natality database for the years 2007 to 2018. We assessed the trends of five pregnancy risk factors/complications among the three largest racial/ethnic groups in the US: non-Hispanic Whites, non-Hispanic Blacks and Hispanics. Other racial/ethnic groups and incomplete records were excluded. The following pregnancy risk factors/complications were examined: hypertensive disorders of pregnancy (HDP; includes gestational hypertension, preeclampsia and eclampsia), chronic hypertension (CH), diabetes mellitus (DM; includes both pre-gestational diabetes and gestational diabetes), advanced maternal age (AMA; defined as a maternal age of at least 35 years at birth) and grand multiparty (GM; defined as 5 or more prior live births).

For every year between 2007 and 2018, we calculated the prevalence of each pregnancy risk factor/complication among each racial/ethnic group. We then compared the prevalence of each pregnancy risk factor/complication across the three racial/ethnic groups. Subsequently, we assessed the temporal trends by comparing the prevalence of each pregnancy risk factor/complication among each racial/ethnic group in 2018 to the prevalence in 2007. Pearson chi-square testing was used for statistical analyses, with statistical significance set as a p-value below 0.05. The results were displayed as odds ratios (OR) with 95% confidence intervals (95% CI) [16]. Institutional review board approval and informed consent were not required because the de-identified data are publicly available through a data use agreement with the National Center for Health Statistics.

## 3. Results

Our study includes a total of 43,890,101 live births that occurred between 2007 and 2018. Non-Hispanic White women accounted for 25,600,706 (58.3%) of these births, Hispanic women for 11,253,199 (25.6%) and non-Hispanic Black women for 7,036,106 (16%).

The number of annual cases and the prevalence of the five risk factors/complications among the three racial/ethnic groups between 2007 and 2018 are presented in Table 1 for HDP, Table 2 for CH, Table 3 for DM, Table 4 for AMA and Table 5 for GM. Odds ratios and 95% confidence intervals comparing the prevalence among each racial/ethnic group in 2018 to that in 2007 were calculated and presented in each table.

A comparison of the prevalence of each pregnancy risk factor/complication across the different racial/ethnic groups in 2018 is displayed in Figure 1. Several differences are noted, with non-Hispanic Blacks demonstrating the highest prevalence of HDP, CH and GM; Hispanics the highest prevalence of DM and non-Hispanic Whites the highest prevalence of AMA (*p* < 0.001).

The prevalence of hypertensive disorders of pregnancy (HDP) was consistently highest among non-Hispanic Black women and consistently lowest among Hispanic women for each and every year between 2007 and 2018 (Table 1). Nonetheless, Hispanics had the largest increase in the prevalence of HDP (OR 2.23, 95% CI 2.2–2.26, *p* < 0.001), followed by non-Hispanic Blacks (OR 1.94, 95% CI 1.91–1.97, *p* < 0.001) and non-Hispanic Whites (OR 1.75, 95% CI 1.74–1.76, *p* < 0.001) (Table 1 and Figure 2).

Similarly, the prevalence of chronic hypertension (CH) was consistently highest among non-Hispanic Black women and consistently lowest among Hispanic women for each and every year between 2007 and 2018 (Table 2). Hispanics, however, had the largest increase in the prevalence of CH between 2007 and 2018 (OR 2.51, 95% CI 2.44–2.6, *p* < 0.001), followed by non-Hispanic Blacks (OR 1.91, 95% CI 1.87–1.95, *p* < 0.001) and non-Hispanic Whites (OR 1.74, 95% CI 1.71–1.77, *p* < 0.001) (Table 2 and Figure 2).

The prevalence of diabetes mellitus (DM) was consistently highest among Hispanic women for all of the years between 2007 and 2018 (Table 3). Interestingly, Hispanics also had the largest increase in the prevalence of DM during this period (OR 1.88, 95% CI 1.85–1.9, *p* < 0.001), followed by non-Hispanic Blacks (OR 1.71, 95% CI 1.68–1.73, *p* < 0.001) and non-Hispanic Whites (OR 1.61, 95% CI 1.59–1.62, *p* < 0.001) (Table 3 and Figure 2).

The prevalence of advanced maternal age (AMA) was consistently highest among non-Hispanic White women, followed by Hispanic and non-Hispanic Black women for all of the years between 2007 and 2018 (Table 4). Nevertheless, the temporal increase in the prevalence of AMA was higher among Hispanics (OR 1.62, 95% CI 1.61–1.64, *p* < 0.001) and non-Hispanic Blacks (OR 1.57, 95% CI 1.55–1.59, *p* < 0.001), compared to non-Hispanic Whites (OR 1.16, 95% CI 1.16–1.17, *p* < 0.001) (Table 4 and Figure 2).

The prevalence of grand multiparity (GM) was consistently highest among non-Hispanic Black women, followed by Hispanic and non-Hispanic White women for all of the years between 2007 and 2018 (Table 5). However, the largest increase in the prevalence of GM was among non-Hispanic Whites (OR 1.29, 95% CI 1.28–1.30, *p* < 0.001), followed by non-Hispanic Blacks (OR 1.21, 95% CI 1.19–1.23, *p* < 0.001) and Hispanics (OR 1.19, 95% CI 1.17–1.2, *p* < 0.001) (Table 5 and Figure 2).

A substantial increase in the prevalence of all pregnancy risk factors/complications was noted between 2007 and 2018, across the different racial/ethnic groups (*p* < 0.001) (Figure 2). Moreover, the documented increases in these risk factors vary among the different racial/ethnic groups. In particular, the highest increases in HDP, CH, DM and AMA were exhibited in Hispanic women, whereas non-Hispanic Black women exhibited the second highest increases in these parameters.

## 4. Discussion

Our analysis detected a significant increase in the prevalence of five major pregnancy risk factors/complications in the US between 2007 and 2018. This was observed among all racial/ethnic groups, i.e., non-Hispanic Whites, non-Hispanic Blacks, and Hispanics. Furthermore, we identified significant racial/ethnic differences in the prevalence, as well as the temporal changes in the prevalence, of these pregnancy risk factors/complications, suggesting that there is a strong association between these risk factors/complications and maternal race/ethnicity. For example, non-Hispanic Black women, who have the highest maternal mortality rates in the US [1,2,3,4,5,6,7], had the highest prevalence of HDP, CH and GM in each and every year between 2007 and 2018, while Hispanic women had the highest prevalence of DM, and non-Hispanic White women had the highest prevalence of AMA.

We illustrate that the temporal increase in the prevalence of both HDP and CH between 2007 and 2018 was highest for Hispanic women. Nevertheless, all racial/ethnic groups demonstrated a considerable increase in these major pregnancy risk factors/complications (OR ranging from 1.75 to 2.23 for HDP and from 1.74 to 2.51 for CH). Moreover, given the historically high prevalence of these risk factors/complications among non-Hispanic Black women, the prevalence remained substantially higher among non-Hispanic Black women compared to Hispanic and non-Hispanic White women. Our results are consistent with previous studies, which have documented a higher prevalence of HDP [8,10,11] and CH [9] among Black women. In addition, non-Hispanic Black women with preeclampsia or eclampsia have been found to have higher complication and mortality rates compared to non-Black women with similar conditions [16]. The substantial increase in the prevalence of HDP and CH over such a relatively short period is extremely alarming as these conditions are associated with significant maternal and neonatal morbidity and mortality, especially among Black women [17]. Potential explanations for the increase in CH and HDP likely include poor dietary habits, sedentary lifestyle and increasing rates of obesity, as well as trends of delaying childbirth, as we have shown. The increase in HDP is also directly related to the increased prevalence of CH and the increasing rates of multiple gestations.

Similar to previous reports, Hispanic ethnicity in our study was associated with the highest risk of DM in pregnancy. Hispanic women also exhibited the highest temporal increase in DM during the 2007–2018 time period (OR 1.88, 95% CI 1.85–1.9, *p* < 0.001). This points to further widening of the DM-related racial/ethnic gap. Higher rates of both pre-existing diabetes and gestational diabetes among Hispanic pregnant women have been reported in several prior studies and being of Hispanic ethnicity is indeed considered a leading risk factor for gestational diabetes [18,19,20,21]. Nonetheless, our data illustrates that DM is not solely a Hispanic problem. A marked increase in DM was also observed in non-Hispanic Black and non-Hispanic White women (OR of 1.71 and 1.61, respectively). Such a striking increase in this relatively short time period solidifies DM as a major and common risk factor/complication among US pregnant women over the last decade. Similarly to the increased risk of HDP, the rise in DM may be attributed to a combination of factors such as sedentary life style, poor dietary habits and increasing obesity rates, as well as to improved screening strategies identifying more patients as affected.

Our findings are consistent with earlier studies that reported a rise in the prevalence of HDP [22], CH [22,23] and DM [24,25] among US pregnant women. For example, Kuklina et al. (2009) reported that hypertensive disorders among delivery hospitalizations increased from 6.72% in 1998 to 8.14% in 2006. Similarly, it has been shown that pre-gestational diabetes among pregnant women increased from 0.76% in 1999 to 1.90% in 2005 and that gestational diabetes increased from 7.1% in 1999 to 7.8% in 2005 [24]. A different study found that the overall prevalence of gestational diabetes increased from 1.9% in 1989–1990 to 4.2% in 2003–2004 [25]. Our analysis expands these prior findings by documenting that on a nationwide scale, there are continuous increasing trends in these pregnancy risk factors/complications. These alarming trends indicate that pregnant women nowadays are more likely to have HDP, CH and DM than in the recent past [10]. Moreover, we provide racial/ethnic specific data regarding these conditions.

Although AMA was consistently more prevalent among non-Hispanic White women for all of the years between 2007 and 2018, the overall increase in the prevalence of AMA was relatively low for this racial/ethnic group (OR 1.16, CI 1.16–1.17, *p* < 0.001) compared to Hispanics (OR 1.62, 95% CI 1.61–1.64, *p* < 0.001) and non-Hispanic Blacks (OR 1.57, CI 1.55–1.59, *p* < 0.001). The increased risk of AMA is most disturbing with regard to non-Hispanic Blacks, as the excess mortality rate of pregnant Black women has been found to be age dependent [3]. The specific etiology as to why non-Hispanic Black and Hispanic women are “closing the gap” on non-Hispanic White women in this respect is unknown. Multiple socio-economic, cultural, medical and educational factors may play a role. In many geographical regions, AMA is more common among a patient population that delays reproduction due to personal or career reasons. This is generally more frequent among patients from middle and high socio-economic status and a higher education level. Thus, improved socio-economic status or access to care for non-Hispanic Blacks and Hispanics, the delay of family planning and improved access to reproductive therapies may all contribute. Unfortunately, ineffective or lack of appropriate contraception counselling or methods may also contribute to pregnancies at advanced maternal age, specifically among high-risk women or those with limited access to medical care.

The increase in the prevalence of GM during the period between 2007 and 2018 was relatively modest for all racial/ethnic groups with a slightly higher trend noted among non-Hispanic Whites (OR 1.29, 95% CI 1.28–1.30, *p* < 0.001). Nevertheless, during our study period, GM remained significantly more prevalent among non-Hispanic Blacks and Hispanics compared to non-Hispanic Whites. Prior studies have reported inconsistent results on the association between GM and pregnancy complications. While certain studies have found GM to be associated with an increased risk of pregnancy complications [26,27,28,29], other studies, primarily in recent years, have suggested that in developed countries with satisfactory prenatal care, GM is not associated with additional risk [30,31,32]. Further research is therefore needed to understand the implications of our results.

The observed racial/ethnic disparities in both the prevalence and the temporal trends in the prevalence of these pregnancy risk factors/complications are not well understood. It is not known why women of Hispanic ethnicity, for example, exhibited the highest increase in the prevalence of HDP, CH, DM and AMA, followed by non-Hispanic Black women. Many possible explanations can potentially explain these observed differences, including differences in socioeconomic status and health insurance coverage [33], discrimination and racism [34], racial segregation [35], differences in environmental factors and neighborhood characteristics [36], genetic and biological differences [37] and differences in the pre-conception health of the mother [33,38]. However, the importance and contribution of each of these potential explanations have yet to be determined.

Our study has several strengths. It expands the evaluation of racial/ethnic differences in pregnancy outcomes by utilizing the largest and most comprehensive source of natality data for the US. The results can therefore be generalized to the entire US population and are not constrained by the characteristics of a specific community. Our analysis is also novel in that we compared not only the prevalence of pregnancy risk factors/complications across different racial/ethnic groups but also the temporal trends in these risk factors/complications over the past decade. To our knowledge, this is the first study to assess the temporal trends in the prevalence of these risk factors/complications among pregnant women from different racial/ethnic groups in the US in the time period leading up to 2018. The results may therefore advance our understanding of the underlying reasons for the observed increase in maternal morbidity and mortality in the US, as well as point to potential explanations for the failed attempts to reduce them over the past decade. As the risk factors/complications that we tested have increased significantly among all major racial/ethnic groups, attention from the medical community and health care policy makers should focus on interventions and educational campaigns that are targeted on their reduction in all racial/ethnic groups. These interventions may include specific programs to facilitate a healthy lifestyle, increased physical activity, better understanding of healthy nutrition and how to implement dietary modifications in a way that is applicable and financially plausible, weight reduction programs, as well as better access to contraception and to pre-conception counseling. Racial/ethnic differences in these risk factors/complications are a reality among US women in the period of 2007–2018. Thus, such interventions should take into account the differences in the culture, dietary habits, education levels and socio-economic status, as well as the language barriers that may impact the access to and success of these mediations in women from different racial/ethnic groups. In addition, further investigation of the influence of these risk factors/complications on pregnancy outcomes in each racial/ethnic group is important in order to understand whether they impact women of different racial/ethnic groups in a similar fashion.

The study also has several limitations that are commonly associated with large registry-based databases. First, the database may contain missing or incorrect records. Second, the CDC database is limited to live births, and data on pregnancies that resulted in voluntary termination or stillbirth due to maternal/fetal complications of pregnancy are not included. This may lead to underrepresentation of the most severe cases. Third, some conditions are all lumped into one variable and cannot be assessed independently in the CDC database. For example, it does not distinguish between pre-gestational diabetes (type 1 or 2) and gestational diabetes. Similarly, the variable of HDP includes all hypertensive disorders of pregnancy (gestational hypertension, mild and severe preeclampsia, and eclampsia) and they cannot be evaluated separately. Finally, recommendations for gestational diabetes screening changed in 2014, and the diagnosis criteria prior to 2014 might have varied between individual practices.

## 5. Conclusions

In conclusion, we report that the prevalence of HDP, CH, DM, AMA and GM increased between 2007 and 2018 among non-Hispanic White, non-Hispanic Black and Hispanic gravidas. Additionally, we detect significant racial/ethnic differences in the overall prevalence, as well as the temporal changes in the prevalence of these risk factors/complications in the period between 2007 and 2018. Further research may be helpful in order to determine the reasons for these concerning increases and racial/ethnic differences and in order to evaluate their contribution to maternal and neonatal morbidity and mortality in the US.

## Figures and Tables

**Figure 1 jcm-09-01414-f001:**
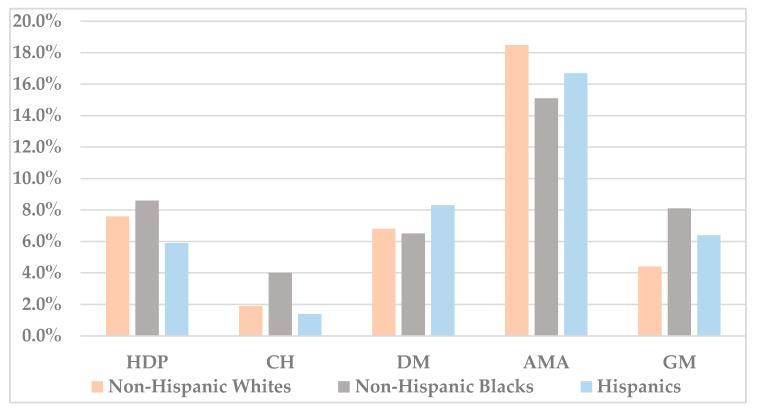
The prevalence of the pregnancy risk factors/complications in 2018, by racial/ethnic group. All of the differences between the racial/ethnic groups are statistically significant (*p* < 0.001).

**Figure 2 jcm-09-01414-f002:**
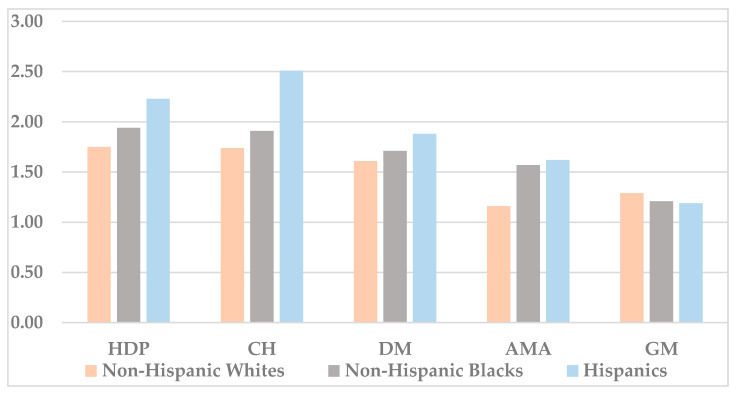
Temporal changes in the prevalence of the pregnancy risk factors/complications between 2007 and 2018, by racial/ethnic group (presented as odds ratios). All of the differences between the racial/ethnic groups are statistically significant (*p* < 0.001).

**Table 1 jcm-09-01414-t001:** Annual prevalence of hypertensive disorders of pregnancy (HDP) for 2007–2018, by racial/ethnic group.

	Non-Hispanic White Women	Non-Hispanic Black Women	Hispanic Women
2007	4.4% (100,077/2,292,263)	4.6% (28,620/618,230)	2.8% (29,110/1,055,699)
2008	4.4% (98,885/2,247,389)	4.8% (29,053/609,494)	2.8% (29,173/1,032,029)
2009	4.6% (101,274/2,195,497)	5.0% (30,160/599,906)	2.9% (28,670/991,422)
2010	4.8% (103,008/2,146,036)	5.4% (31,023/577,815)	3.2% (30,023/936,964)
2011	4.8% (102,156/2,127,389)	5.5% (31,263/570,370)	3.3% (30,222/911,989)
2012	5.0% (106,104/2,120,779)	5.7% (32,599/575,253)	3.6% (32,365/903,035)
2013	5.2% (110,201/2,117,564)	5.9% (33,829/576,509)	3.8% (33,778/896,734)
2014	5.5% (116,615/2,137,961)	6.2% (36,097/582,976)	4.1% (37,052/909,071)
2015	5.9% (125,784/2,118,909)	7.0% (40,706/583,916)	4.6% (41,970/920,204)
2016	6.4% (133,241/2,084,603)	7.4% (42,656/579,334)	4.9% (44,618/915,506)
2017	6.9% (139,582/2,024,194)	7.8% (45,606/585,170)	5.3% (47,609/896,593)
2018	7.6% (151,962/1,988,122)	8.6% (49,665/577,133)	5.9% (52,574/883,953)
2007–2018 Diff	OR: 1.75 (95% CI: 1.74–1.76)	OR: 1.94 (95% CI: 1.91–1.97)	OR: 2.23 (95% CI: 2.20–2.26)
*p*-value	<0.001	<0.001	<0.001

**Table 2 jcm-09-01414-t002:** Annual prevalence of chronic hypertension (CH) for 2007–2018, by racial/ethnic group.

	Non-Hispanic White Women	Non-Hispanic Black Women	Hispanic Women
2007	1.1% (25,589/2,292,263)	2.2% (13,340/618,230)	0.5% (5721/1,055,699)
2008	1.2% (26,545/2,247,389)	2.4% (14,470/609,494)	0.6% (6321/1,032,029)
2009	1.2% (26,949/2,195,497)	2.6% (15,452/599,906)	0.7% (6708/991,422)
2010	1.3% (27,715/2,146,036)	2.8% (16,258/577,815)	0.7% (6781/936,964)
2011	1.4% (29,039/2,127,389)	2.9% (16,790/570,370)	0.8% (7238/911,989)
2012	1.4% (29,192/2,120,779)	3.1% (17,584/575,253)	0.8% (7568/903,035)
2013	1.4% (30,291/2,117,564)	3.1% (17,908/576,509)	0.9% (8486/896,734)
2014	1.5% (31,980/2,137,961)	3.2% (18,509/582,976)	1.0% (8987/909,071)
2015	1.5% (32,195/2,118,909)	3.4% (19,916/583,916)	1.0% (9265/920,204)
2016	1.6% (33,782/2,084,603)	3.5% (20,418/579,334)	1.1% (10,074/915,506)
2017	1.8% (35,667/2,024,194)	3.8% (22,160/585,170)	1.2% (10,906/896,593)
2018	1.9% (38,605/1,988,122)	4.0% (23,296/577,133)	1.4% (11,948/883,953)
2007–2018 Diff	OR: 1.74 (95% CI: 1.71–1.77)	OR: 1.91 (95% CI: 1.87–1.95)	OR: 2.51 (95% CI: 2.44–2.60)
*p*-value	<0.001	<0.001	<0.001

**Table 3 jcm-09-01414-t003:** Annual prevalence of diabetes mellitus (DM) for 2007–2018, by racial/ethnic group.

	Non-Hispanic White Women	Non-Hispanic Black Women	Hispanic Women
2007	4.2% (97,013/2,292,263)	3.9% (24,095/618,230)	4.6% (48,737/1,055,699)
2008	4.3% (95,850/2,247,389)	4.0% (24,594/609,494)	4.8% (49,044/1,032,029)
2009	4.4% (97,210/2,195,497)	4.2% (24,966/599,906)	5.0% (49,182/991,422)
2010	4.7% (101,252/2,146,036)	4.5% (25,794/577,815)	5.2% (48,958/936,964)
2011	5.1% (108,680/2,127,389)	4.9% (28,124/570,370)	5.8% (53,195/911,989)
2012	5.3% (113,450/2,120,779)	5.2% (30,056/575,253)	6.3% (57,285/903,035)
2013	5.4% (114,967/2,117,564)	5.4% (30,898/576,509)	6.5% (58,230/896,734)
2014	5.6% (119,982/2,137,961)	5.6% (32,393/582,976)	6.9% (62,745/909,071)
2015	5.8% (123,489/2,118,909)	5.7% (33,533/583,916)	7.1% (65,701/920,204)
2016	6.0% (125,616/2,084,603)	6.0% (34,751/579,334)	7.5% (68,840/915,506)
2017	6.5% (131,110/2,024,194)	6.3% (36,642/585,170)	8.0% (71,302/896,593)
2018	6.8% (135,087/1,988,122)	6.5% (37,328/577,133)	8.3% (73,582/883,953)
2007–2018 Diff	OR: 1.61 (95% CI: 1.59–1.62)	OR: 1.71 (95% CI: 1.68–1.73)	OR: 1.88 (95% CI: 1.85–1.90)
*p*-value	<0.001	<0.001	<0.001

**Table 4 jcm-09-01414-t004:** Annual prevalence of advanced maternal age (AMA) for 2007–2018, by racial/ethnic group.

	Non-Hispanic White Women	Non-Hispanic Black Women	Hispanic Women
2007	15.9% (364,969/2,292,263)	10.2% (62,997/618,230)	11.0% (116,368/1,055,699)
2008	15.6% (349,939/2,247,389)	10.2% (62,190/609,494)	11.5% (118,602/1,032,029)
2009	15.3% (335,558/2,195,497)	10.3% (62,082/599,906)	12.0% (119,110/991,422)
2010	15.2% (326,424/2,146,036)	10.7% (61,943/577,815)	12.7% (119,385/936,964)
2011	15.2% (322,367/2,127,389)	11.0% (62,741/570,370)	13.3% (121,481/911,989)
2012	15.2% (322,620/2,120,779)	11.3% (64,966/575,253)	13.7% (124,099/903,035)
2013	15.5% (328,632/2,117,564)	11.7% (67,639/576,509)	14.3% (127,799/896,734)
2014	15.8% (338,810/2,137,961)	12.3% (71,980/582,976)	14.7% (133,905/909,071)
2015	16.4% (348,513/2,118,909)	13.0% (75,838/583,916)	15.2% (140,012/920,204)
2016	17.1% (356,242/2,084,603)	13.8% (79,816/579,334)	15.9% (145,269/915,506)
2017	17.8% (359,651/2,024,194)	14.3% (83,836/585,170)	16.3% (146,552/896,593)
2018	18.5% (368,253/1,988,122)	15.1% (87,229/577,133)	16.7% (148,040/883,953)
2007-2018 Diff	OR: 1.16(95% CI: 1.16–1.17)	OR: 1.57 (95% CI: 1.55–1.59)	OR: 1.62 (95% CI: 1.61–1.64)
*p*-value	<0.001	<0.001	<0.001

**Table 5 jcm-09-01414-t005:** Annual prevalence of grand multiparity (GM) for 2007–2018, by racial/ethnic group.

	Non-Hispanic White Women	Non-Hispanic Black Women	Hispanic Women
2007	3.4% (77,879/2,292,263)	6.8% (41,850/618,230)	5.5% (57,661/1,055,699)
2008	3.5% (77,902/2,247,389)	6.8% (41,369/609,494)	5.6% (57,833/1,032,029)
2009	3.6% (78,593/2,195,497)	6.9% (41,542/599,906)	5.8% (57,035/991,422)
2010	3.7% (78,660/2,146,036)	7.0% (40,224/577,815)	6.0% (56,146/936,964)
2011	3.7% (79,026/2,127,389)	7.1% (40,548/570,370)	6.1% (55,178/911,989)
2012	3.8% (80,533/2,120,779)	7.1% (41,024/575,253)	6.2% (55,892/903,035)
2013	3.9% (81,712/2,117,564)	7.3% (42,071/576,509)	6.3% (56,780/896,734)
2014	4.0% (84,482/2,137,961)	7.5% (43,618/582,976)	6.4% (57,872/909,071)
2015	4.1% (85,883/2,118,909)	7.6% (44,380/583,916)	6.4% (58,583/920,204)
2016	4.2% (86,788/2,084,603)	7.8% (45,135/579,334)	6.5% (59,540/915,506)
2017	4.3% (87,344/2,024,194)	8.0% (47,091/585,170)	6.5% (58,379/896,593)
2018	4.4% (87,047/1,988,122)	8.1% (46,653/577,133)	6.4% (56,692/883,953)
2007–2018 Diff	OR: 1.29 (95% CI: 1.28–1.30)	OR: 1.21 (95% CI: 1.19–1.23)	OR: 1.19 (95% CI: 1.17–1.20)
*p*-value	<0.001	<0.001	<0.001

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
