# Peer review of "Racial Disparity in Pregnancy Risks and Complications in the US: Temporal Changes during 2007–2018"

_jcm, 2020, doi:10.3390/jcm9051414_

Round 1
Reviewer 1 Report
Thank you for the opportunity to read this interesting manuscript which attempts to ethnicity differences in incremental rates of risk factors associated with adverse pregnancy outcomes. The strengths of the study are the size of the dataset, length of period studied and ethnic diversity of the cohort.
The authors do not overstate or speculate inappropriately about their findings, and they recognise the limitations of the dataset which is commendable.
I have a few suggestion to improve the manuscript.
I am unclear where the statistical comparisons between ethnicity groups are presented other than in the prose, and suggest this could be added to Figure 2.
The authors may have already considered exploring the relationship between the risk factors – CHT, AMA, pre-existing DM and HDP. Are there any important ethnicity differences identified? Which risk factor is strongest in multivariate analysis?
The authors mention that ethnicity related factors may explain why targeted interventions may have failed. This idea could be developed further, as it raises the most important potential impact of the findings.
Author Response
May 06 2020
Re: Manuscript ID: jcm-800425
Title: Racial Disparity in Pregnancy Risks and Complications in the US: Temporal Changes
during 2007-2018
Authors: Eran Bornstein *, Yael Eliner, Frank A Chervenak, Amos Grünebaum
Dear Ms. Blagojevic,
Please convey our deep appreciation to both reviewers for their thoughtful comments. We have revised the manuscript as suggested by the reviewers. An updated version is attached with the revisions in “Track Changes”. Our responses to the reviewers’ comments are as follows:
Reviewer 1:
- I am unclear where the statistical comparisons between ethnicity groups are presented other than in the prose, and suggest this could be added to Figure 2.
Response: As suggested, we added a footnote to both Figures (line 88 and 141) stating: “… * All of the differences between the racial/ethnic groups are statistically significant (p<0.001)”.
- The authors may have already considered exploring the relationship between the risk factors – CHT, AMA, pre-existing DM and HDP. Are there any important ethnicity differences identified? Which risk factor is strongest in multivariate analysis?
Response: We thank the reviewer for this comment. Multivariate analysis indeed may help determining which risk factor is the strongest leading to obstetric morbidity and mortality. Nevertheless, this would be a different objective from our study which was aimed at comparing the prevalence and the temporal changes in the prevalence of HDP, CH, DM, AMA and GM across the three racial/ethnic groups over specific time period. We think this is a great suggestion for subsequent research assessing the relationship between these risk factors leading to poor outcome in the different ethnicities. We added a sentence to the discussion (lines 252-254) to reflect this idea: “…In addition, further investigation of the influence of these factors on pregnancy outcomes in each racial/ethnic group is important to understand whether they impact women of different race/ethnic group in a similar fashion”.
- The authors mention that ethnicity related factors may explain why targeted interventions may have failed. This idea could be developed further, as it raises the most important potential impact of the findings.
Response: As suggested, we expanded the discussion to further develop this topic and added the following to lines 241-254: “…As the risk factors and complications we tested have increased significantly in all major racial/ethnic groups, major attention from the medical community and health care policy makers should focus on interventions and educational campaigns targeted on their reduction in all racial/ethnic groups. These interventions may include specific programs to facilitate healthy life style, increased physical activity, better understanding of healthy nutrition and how to implement dietary modifications in a way that is applicable and financially plausible, weight reduction programs, as well as better access to contraception and to pre-conception counseling. Racial/ethnic differences in these risk factors and complications are a reality among US women in 2007-2018. Thus, such interventions should take into account the cultural, dietary habits, educational level, socio-economic gaps, and language barriers that may impact the access to and success of these mediations in women from different racial/ethnic groups. In addition, further investigation of the influence of these factors on pregnancy outcomes in each racial/ethnic group is important to understand whether they impact women of different race/ethnic group in a similar fashion”.
Additionally, a few minor editing corrections were made throughout the manuscript. They are all highlighted in “Track Changes”.
Thank you again for reviewing our manuscript. We have responded to all the comments made by the reviewers, made the requested changes in the manuscript and hope that you will find the revised manuscript suitable for publication.
Sincerely,
Eran Bornstein, MD
Vice Chair, OBGYN
Director, Maternal Fetal Medicine
Lenox-Hill Hospital, NorthWell

Reviewer 2 Report
This retrospective cohort study evaluates temporal trends of common pregnancy risk factors/complications in the American pregnant population in the study period 2007-2018. The study provides a comprehensive insight into the ethnical distribution of the studied parameters which can be generalized to the US population. The study found increased prevalence of all studied risk factors (gestational hypertensive disorders, chronic hypertension, diabetes mellitus, advanced maternal age and grand multiparity). The main strength of this study is the large study population. I would suggest the following:
- In the introduction section please provide maternal mortality rates for all three ethnical groups
- In the methods section does the database enable separate analysis of the trends of gestational hypertension, preeclampsia and eclampsia within gestational hypertensive disorders group?
- In the discussion section (page 7, paragraph 2) the authors address possible reasons for the observed trends. I would suggest a somewhat more detailed discussion for each complication in each respective paragraph (page 6, paragraph 2-possible reasons for the increase in the prevalence of hypertensive disorders; page 6, paragraph 3-possible reasons for the increase of gestational diabetes…)
Author Response
May 06 2020
Re: Manuscript ID: jcm-800425
Title: Racial Disparity in Pregnancy Risks and Complications in the US: Temporal Changes
during 2007-2018
Authors: Eran Bornstein *, Yael Eliner, Frank A Chervenak, Amos Grünebaum
Dear Ms. Blagojevic,
Please convey our deep appreciation to both reviewers for their thoughtful comments. We have revised the manuscript as suggested by the reviewers. An updated version is attached with the revisions in “Track Changes”. Our responses to the reviewers’ comments are as follows:
Reviewer 2:
- In the introduction section please provide maternal mortality rates for all three ethnical groups.
Response: As suggested, maternal mortality rates for each racial/ethnic group in 2018 were added in lines 33-35 as well as a reference for the source: “…For example, in 2018, the US maternal mortality rate for Black women was 37.1 per 100,000 live births, whereas the rates for White and Hispanic women were 14.7 and 11.8 per 100,000 live births, respectively [7]”.
- In the methods section does the database enable separate analysis of the trends of gestational hypertension, preeclampsia and eclampsia within gestational hypertensive disorders group?
Response: Unfortunately, the CDC natality data does not distinguish between gestational hypertension, preeclampsia and eclampsia which are lumped together so separate analysis is not possible. We added a clarification regarding this issue to the paragraph discussing the limitations of our study in lines 260-264: “…Third, some conditions are all lumped into one variable and cannot be assessed independently in the CDC database. For example, it does not distinguish between pre-gestational diabetes (type 1 or 2) and gestational diabetes. Similarly, the variable of HDP includes all hypertensive disorders of pregnancy (gestational hypertension, mild and severe preeclampsia, and eclampsia) and they cannot be evaluated separately.”.
- In the discussion section (page 7, paragraph 2) the authors address possible reasons for the observed trends. I would suggest a somewhat more detailed discussion for each complication in each respective paragraph (page 6, paragraph 2-possible reasons for the increase in the prevalence of hypertensive disorders; page 6, paragraph 3-possible reasons for the increase of gestational diabetes…)
Response: As suggested, we expanded the discussion on potential reasons for the increase in each respective complication.
Comments regarding potential reasons for increase in HDP and CH in lines 165-168: “…Potential explanations for the increase in CH and HDP likely includes both poor dietary habits, sedentary lifestyle and increasing rates of obesity as well as trends of delaying childbirth as we have shown. The increase in HDP is also directly related to the increased prevalence of CH and by increasing rates of multiple gestations”.
Comments regarding the potential reasons for increase in DM were added in lines 178-181: “…Similarly to the increased risk of HDP, the rise in DM may be attributed to a combination of factors such as sedentary life style, poor dietary habits and increasing obesity rates, as well as to improved screening strategies identifying more patients as affected”.
A discussion of potential reasons for the increase in AMA was added in lines 198-206: “The specific etiology as to why non-Hispanic Black and Hispanic women are “closing the gap” on non-Hispanic White women in this respect is unknown. Multiple socio-economic, cultural, medical and educational factors may play a role. In many geographical regions, AMA is more common in patient population that delays reproduction due to personal or career reasons. This is generally more frequent in patients from middle and high socio economic status and a higher education level. Thus, improved socioeconomic status or access to care for non-Hispanic Blacks and Hispanics, the delay of family planning and improved access to reproductive therapies may all contribute. Unfortunately, ineffective or lack of appropriate contraception counselling or methods may also contribute to pregnancies at advanced maternal age specifically in high risk women or those with limited access to medical care”.
Following these changes, subsequent changes were made in lines 225-227 to avoid repetitiveness.
Additionally, a few minor editing corrections were made throughout the manuscript. They are all highlighted in “Track Changes”.
Thank you again for reviewing our manuscript. We have responded to all the comments made by the reviewers, made the requested changes in the manuscript and hope that you will find the revised manuscript suitable for publication.
Sincerely,
Eran Bornstein, MD
Vice Chair, OBGYN
Director, Maternal Fetal Medicine
Lenox-Hill Hospital, NorthWell
